# Application of Solution Calorimetry to Determining the Fusion Enthalpy of an Arylaliphatic Compound at 298.15 K: *n*-Octadecanophenone

Mikhail I. Yagofarov *[ID], Ilya S. Balakhontsev, Andrey A. Sokolov [ID] and Boris N. Solomonov

Department of Physical Chemistry, Kazan Federal University, Kremlevskaya Str. 18, 420008 Kazan, Russia
* Correspondence: miiyagofarov@kpfu.ru

**Abstract:** Evaluating the temperature dependence of the fusion enthalpy is no trivial task, as any compound melts at a unique temperature. At the same time, knowledge of the fusion enthalpies under some common conditions, particularly at the reference temperature of 298.15 K, would substantially facilitate the comparative analysis and development of the predictive schemes. In this work, we continue our investigations of the temperature dependence of the fusion enthalpy of organic non-electrolytes using solution calorimetry. As an object of study, *n*-octadecanophenone, an arylaliphatic compound was chosen. The solvent appropriate for evaluating the fusion enthalpy at 298.15 K from the solution enthalpy of crystal was selected: *p*-xylene. The heat capacity and fusion enthalpy at the melting temperature were measured by differential scanning calorimetry to derive the fusion enthalpy at 298.15 K from the Kirchhoff's law of Thermochemistry. An agreement between the independently determined values was demonstrated. This particular result opens a perspective for further studies of the fusion thermochemistry of arylaliphatic compounds at 298.15 K by solution calorimetry.

**Keywords:** solution calorimetry; fusion enthalpy; differential scanning calorimetry; alkanophenones

## 1. Introduction

The studies of phase transition thermodynamics of organic non-electrolytes have more than a 150-year-old history. A lot of efforts have been paid to develop increasingly precise experimental techniques [1], as well as predictive methods. Quantitative structure-property relationships proposed so far to characterize the fusion process, particularly its enthalpy, remain relatively less accurate [2–7], compared with the vaporization and sublimation [7–12]. Among the obstacles to establishing the relationships of the fusion enthalpy with the molecular structure and descriptors, one can distinguish the fact that it is measured at the melting point ($T_m$), which is unique for each compound. This impedes the comparative analysis, in contrast with the vaporization and sublimation processes, whose thermodynamic parameters can be determined under the common conditions (e.g., 298.15 K) for numerous systems. Adjustment of the fusion enthalpies ($\Delta_{cr}^{l}H$) to 298.15 K would require knowledge of its temperature dependence. The temperature dependence of the fusion enthalpy is also of interest when applying the ideal solubility equation [13–17], and analyzing nucleation and crystallization kinetics [18,19].

$\Delta_{cr}^{l}H$ at 298.15 K can be found from Kirchhoff's law of Thermochemistry (Equation (1)):

$$\Delta_{cr}^{l}H(298.15\text{ K}) = \Delta_{cr}^{l}H(T_m) + \int_{T_m}^{298.15\text{ K}} \Delta_{cr}^{l}C_{p,m}dT \tag{1}$$

where $\Delta_{cr}^{l}C_{p,m}$ is the heat capacity change on melting. Generally speaking, it also depends on temperature and this dependence can sometimes significantly contribute to $\Delta_{cr}^{l}H(298.15\text{ K})$ [20].

Experimental determination of this value below $T_m$ would require supercooling the melt, which is rarely possible.

In a recent cycle of works [21,22] we proposed a method for determining the fusion enthalpies of organic non-electrolytes based on solution calorimetry, based on Hess's law (Equation (2)):

$$\Delta_{soln}H(\text{cr, 298.15 K}) = \Delta_{cr}^{l}H(\text{298.15 K}) + \Delta_{soln}H(\text{l, 298.15 K}) \tag{2}$$

According to Equation (2), the solution enthalpy of the crystal in a certain solvent $\Delta_{soln}H(\text{cr, 298.15 K})$ is equal to a sum of $\Delta_{cr}^{l}H(\text{298.15 K})$ and the solution enthalpy of the quasi-equilibrium melt in the same solvent at 298.15 K. In many solute-solvent systems (e.g., aromatic compounds incapable of self-association in benzene [21] or alkanes in heptane [23]) the latter is nearly 0, so $\Delta_{cr}^{l}H(\text{298.15 K})$ can be found directly from $\Delta_{soln}H(\text{cr, 298.15 K})$.

Combining Equations (1) and (2), we independently tracked the temperature dependence of the supercooled liquid heat capacity between 298.15 K and $T_m$:

$$\Delta_{cr}^{l}H(T_m) + \int_{T_m}^{298.15\text{ K}} \Delta_{cr}^{l}C_{p,m}dT = \Delta_{soln}H(\text{cr, 298.15 K}) - \Delta_{soln}H(\text{l, 298.15 K}) \tag{3}$$

Analyzing the relationship between the enthalpies of fusion and solution, we established that an agreement between the left and right sides of Equation (3) is achieved when the temperature dependence of the heat capacity of the liquid determined above $T_m$ is extrapolated down to 298.15 K as a linear function of temperature [21].

Equation (2) provides a useful alternative to Equation (1), replacing a laborious procedure of heat capacity measurement with solution calorimetry. However, it requires searching for the solvent, in which $\Delta_{soln}H(\text{l, 298.15 K})$ can be accurately evaluated based on the molecular structure. Previously we mainly focused on aromatic (both hydrogen-bonded [22] and non-hydrogen-bonded [21]) and aliphatic [23] systems but did not concern alkylaromatic compounds. For alkylarenes, the solution enthalpy in benzene significantly grows with the chain length increase, while the solution enthalpy in heptane can be significantly influenced by the presence of any substituent, except for the alkyl group. In this work we tested if *p*-xylene, which contains both aromatic core and alkyl groups, can be used as an "athermal" solvent for a long-chain alkylaromatic solute, *n*-octadecanophenone, to determine its fusion enthalpy at 298.15 K.

## 2. Materials and Methods

### 2.1. Materials

*p*-Xylene (CAS № 106-42-3, $C_8H_{10}$, Acros, Thermo Fisher Scientific, Waltham, MA, USA), *n*-octanophenone (CAS № 1674-37-9, $C_{14}H_{20}O$, TCI Chemicals, Tokyo, Japan), and *n*-octadecanophenone (CAS № 6786-36-3, $C_{24}H_{40}O$, Alfa Aesar, Haverhill, MA, USA) were of commercial origin with a purity more than 0.99 (mole fraction), as it was stated in the certificate of analysis (determined by gas chromatography). Water content in *p*-xylene was determined by Fischer titration and equaled 0.01% (mole fraction). Before the measurements, *n*-octanophenone and *n*-octadecanophenone were dried *in vacuo* to remove any traces of moisture.

The samples were used without further purification.

### 2.2. Differential Scanning Calorimetry

The specific heat capacity of crystal, enthalpy, and temperature of fusion of *n*-octadecanophenone were measured using DSC 8500 (Perkin Elmer, Waltham, MA, USA). Prior to the experiment, aluminum crucibles were annealed at 200 °C. DSC was calibrated according to the manufacturer's recommendation using the standard samples of Indium and Zinc. Each value (onset temperature and area of the peak) was determined three times. The reproducibilities of heat flow and temperature calibration (0.95 level of confidence,

coverage factor 2.0) were equal to 1% and 0.1 K, respectively. The correctness of the determination of the heat capacity and heat effects was checked as previously [24]. The agreement with the reference values was within 2%.

The samples were placed in a 50 μL aluminum crucible in an inert atmosphere. Two samples were studied to determine each value. Experiments were performed in a nitrogen dynamic atmosphere (30 mL min$^{-1}$) with a heating/cooling rate of 5 K min$^{-1}$. Two cycles of "heating–cooling" from room temperature to a temperature 40 K higher than the melting point were carried out to determine the enthalpy and temperature of fusion. The samples crystallized completely on cooling. Experimental results from DSC measurements are presented in Table 1. An exemplary melting peak of *n*-octadecanophenone obtained by DSC is shown in Figure 1. An agreement between the fusion enthalpies obtained in each experiment was within a typical reproducibility of the DSC technique (2–3%).

**Table 1.** The enthalpies and temperatures of fusion of *n*-octadecanophenone determined in this work by DSC at 0.1 MPa [a].

| $m$/mg | $T_m$/K | $\Delta_{cr}^{l}H(T_m)/(\text{kJ}\cdot\text{mol}^{-1})$ |
|---|---|---|
| 8.54 | 337.6 | 70.65 |
| 8.54 | 337.6 | 70.84 |
| 10.31 | 337.5 | 71.82 |
| 10.31 | 337.5 | 71.95 |
| Average [b] | 337.6 ± 0.1 | 71.3 ± 2.1 |

[a] The standard uncertainty $u(p)$ = 5 kPa. [b] The uncertainty includes the standard deviation of the mean and the standard calibration uncertainty both multiplied by the coverage factor $k \approx 2.0$ (expanded uncertainty of the mean U; 0.95 level of confidence).

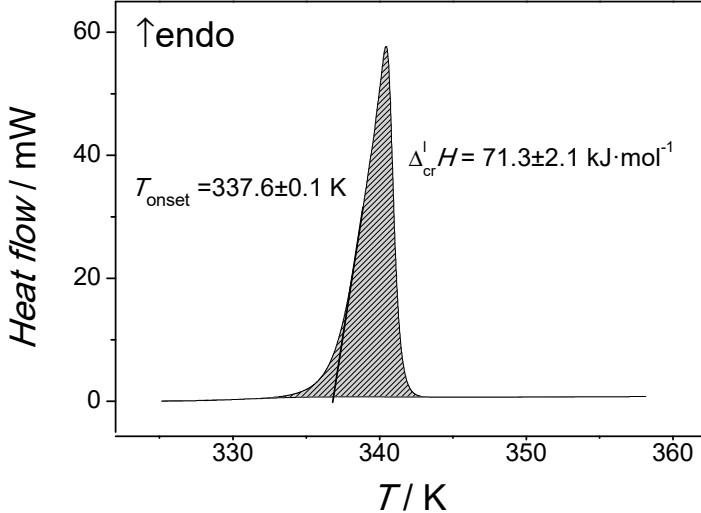

**Figure 1.** Melting peak of *n*-octadecanophenone obtained by DSC in this work.

The measurement of specific heat capacity was performed according to the isothermal step method as previously [25]. The procedure included three steps. First, the baseline was determined using empty crucibles. Then, using the baseline, a standard sample (sapphire) with a weight of 33.79 mg and a sample of *n*-octadecanophenone were each measured in the same crucible. This procedure was repeated twice for two samples of *n*-octadecanophenone, whose fusion enthalpies were measured. The temperature program included a dynamic segment between 320 K and 325 K and two isothermal segments at 320 K and 325 K. The resulting heat capacity of solid *n*-octadecanophenone at 322.5 K was 622 ± 18 J·K$^{-1}$·mol$^{-1}$ (the uncertainty includes the standard deviation of the mean and the standard calibration uncertainty both multiplied by the coverage factor $k \approx 2.0$ (expanded uncertainty of the mean U; 0.95 level of confidence)).

### 2.3. Solution Calorimetry

The solution enthalpies of *n*-octadecanophenone and *n*-octanophenone in *p*-xylene were measured at 298.15 K in the concentration range from 1.39 to 6.37 mmol kg$^{-1}$ using TAM III precision solution calorimeter (TA Instruments, New Castle, DE, USA). Crystal *n*-octadecanophenone was dissolved by breaking a glass ampule filled with ~50 mg of the studied sample in a glass cell containing 90 mL of the pure solvent. Liquid *n*-octanophenone was injected in ~50 μL portions using an electronically operated syringe equipped with a long gold cannula immersed in the solvent. The details of the solution calorimetry procedure have been described elsewhere [26]. The conditions corresponding to an infinite dilution, which was confirmed by an absence of the concentration dependence of the solution enthalpy. The experimental values are provided in Table 2.

**Table 2.** The solution enthalpies of phenones in *p*-xylene measured in this work at 298.15 K and 0.1 MPa [a].

| Compound | *m*/mg [b] | *b*/(mmol·kg$^{-1}$) [c] | $\Delta_{soln}H$/(kJ·mol$^{-1}$) [d] |
|---|---|---|---|
| *n*-Octanophenone | 49.8 | 3.15 | −0.28 |
|  | 50.6 | 3.20 | −0.48 |
|  | 51.0 | 6.37 | −0.34 |
|  | 49.7 | 6.34 | −0.51 |
|  | **Average** [e] |  | −0.40 ± 0.11 |
| *n*-Octadecanophenone | 37.2 | 1.39 | 71.54 |
|  | 48.9 | 1.83 | 70.78 |
|  | 43.8 | 3.03 | 71.07 |
|  | 43.9 | 3.48 | 70.72 |
|  | **Average** [e] |  | 71.0 ± 0.5 |

[a] Standard uncertainties *u* are *u(T)* = 0.01 K, *u(p)* = 5 kPa. [b] Mass of the solute sample which was added in each dissolution experiment. [c] Molality of solute in solution after experiments. Standard uncertainty *u(b)* = 0.01 mmol·kg$^{-1}$. [d] Enthalpy of solution of each experiment. [e] Average enthalpy of solution. The uncertainty includes the standard deviation of the mean and the standard calibration uncertainty both multiplied by the coverage factor $k \approx 2.0$ (expanded uncertainty of the mean U; 0.95 level of confidence).

## 3. Results

In Ref. [27] the molar heat capacity of liquid *n*-octadecanophenone was determined as a function of temperature between 373 and 418 K with an uncertainty of 2% (Equation (4)):

$$C_{p,m}/(\text{J·K}^{-1}\text{·mol}^{-1}) = 297.5 + 1.238 \cdot (T/K) \tag{4}$$

In this work, $C_{p,m}(\text{cr}, 322.5\text{ K}) = 622 \pm 18$ J·K$^{-1}$·mol$^{-1}$ was determined. The extrapolated $C_{p,m}(\text{l}, 322.5\text{ K})$ value equals $697 \pm 30$ J·K$^{-1}$·mol$^{-1}$ (uncertainty evaluated according to Ref. [20]). Thus, $\Delta_{cr}^{l}C_{p,m}(322.5\text{ K}) = 75 \pm 35$ J·K$^{-1}$·mol$^{-1}$. Within the temperature range of Equation (4) (298.15–337.6 K), it is reasonable to assume that $\Delta_{cr}^{l}C_{p,m}$ slightly depends on temperature. Then $\int_{T_m}^{298.15\,K} \Delta_{cr}^{l}C_{p,m}dT = \Delta_{cr}^{l}C_{p,m} \cdot (298.15\text{ K} - T_m) = -3.0 \pm 1.4$ kJ·mol$^{-1}$. Therefore, from Equation (1) one can obtain $\Delta_{cr}^{l}H(298.15\text{ K}) = 68.3 \pm 2.5$ kJ·mol$^{-1}$.

Evaluation of $\Delta_{cr}^{l}H(298.15\text{ K})$ is also possible using the experimental value of $\Delta_{soln}H$ of crystal *n*-octadecanophenone ($71.0 \pm 0.5$ kJ·mol$^{-1}$) and Equation (2). The solution enthalpies of liquid alkanes in *p*-xylene are notably less than in benzene, which has been previously used as a solvent for aromatic compounds. For example, $\Delta_{soln}H(\text{l}, 298.15\text{ K})$ of hexadecane in benzene is equal to 11.3 kJ·mol$^{-1}$ [28] and in *p*-xylene to 3.1 kJ·mol$^{-1}$ [29]. Those of heptane equal 5.6 [28] and 1.4 kJ·mol$^{-1}$ [29], respectively. $\Delta_{soln}H(\text{l}, 298.15\text{ K})$ of *n*-octanophenone in *p*-xylene determined in this work equaled −0.4 kJ·mol$^{-1}$. Thus, it is reasonable to assume that $\Delta_{soln}H(\text{l}, 298.15\text{ K})$ of *n*-octadecanophenone in *p*-xylene is in the range from −0.4 to 3.1 kJ·mol$^{-1}$, or $1.4 \pm 1.8$ kJ·mol$^{-1}$, which is not quite wide, taking in mind the absolute values of $\Delta_{soln}H(\text{cr}, 298.15\text{ K})$ and $\Delta_{cr}^{l}H(T_m)$. Further elaboration of this value is possible if additional measurements of the solution enthalpies of alkylarenes is

performed to understand the regularities met in this series. Then, from Equation (2), one obtains $\Delta_{cr}^{l}H(298.15 \text{ K}) = 69.6 \pm 1.9 \text{ kJ}\cdot\text{mol}^{-1}$.

## 4. Discussion

The above-obtained $\Delta_{cr}^{l}H(298.15 \text{ K})$ values of 68.3 ± 2.5 and 69.6 ± 1.9 kJ·mol$^{-1}$ agree within the limits of the propagated errors. Such an agreement confirms the validity of the assumptions made during the $\Delta_{soln}H(\text{l, 298.15 K})$ evaluation. Thus, the fusion enthalpies of alkylarenes and their derivatives can actually be determined using solution calorimetry and *p*-xylene as an "athermal" solvent. It is worth noting that, due to an extrapolation uncertainty, the error of the enthalpy correction to 298.15 K according to Kirchhoff's law may attain ~50% of its value. In this paper, this error was comparable with the fusion and solution enthalpies uncertainties. However, when the temperature range of adjustment and heat capacity integral increase, its contribution can become crucial. This highlights the advantages of the solution calorimetry approach, which enables to evaluation of the fusion enthalpy directly at 298.15 K, with an uncertainty independent of the melting temperature and tendency to supercooling.

Further quantification of the regularities, especially tracking the effects of the chain length, branching, and substituents on $\Delta_{soln}H(\text{l, 298.15 K})$ in *p*-xylene is anticipated to obtain more accurate results for a wider range of compounds. This finding echoes with the previously denoted "molecule-additivity" of the vaporization enthalpies of alkylaromatic compounds [30], which can be associated with instantaneous nanoheterogeneities in such systems, i.e., aromatic-aromatic and aliphatic-aliphatic interactions are more frequent than aromatic-aliphatic ones. Such distinction can be enough to minimize the significant endothermic effects met when studying benzene-alkanes systems under infinite dilution conditions [28].

**Author Contributions:** Conceptualization, M.I.Y.; methodology, M.I.Y.; investigation, I.S.B. and A.A.S.; writing—original draft preparation, M.I.Y.; writing—review and editing, B.N.S. All authors have read and agreed to the published version of the manuscript.

**Funding:** This paper has been supported by the Kazan Federal University Strategic Academic Leadership Program ('PRIORITY-2030').

**Data Availability Statement:** Not applicable.

**Conflicts of Interest:** The authors declare no conflict of interest.

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
