# Peer review of "Application of Solution Calorimetry to Determining the Fusion Enthalpy of an Arylaliphatic Compound at 298.15 K: n-Octadecanophenone"

_liquids, doi:10.3390/liquids3010001_

Round 1

Reviewer 1 Report

The manuscript presents a continuation of work designed to determine enthalpies of fusion at temperatures different from the melting point, using enthalpies of solution.

The experimental work seems to have been carefully carried out and, reassuringly, the uncertainty budgets provided are consistent with what one would expect for such measurements.

Certainly the work warrants publication.

I would like to have seen a description of the storage and handling of the chemicals.

In the text, units are sometimes written using the mid-dot as separator and other times using the space. One of these must be the style of the journal.

Author Response

We thank the reviewers for their attention and valuable comments that helped to improve the article. The changes in the manuscript were made in review mode and can be seen as a red underlined text. The responses to each comment are provided below.

Reviewer #1

The manuscript presents a continuation of work designed to determine enthalpies of fusion at temperatures different from the melting point, using enthalpies of solution.

The experimental work seems to have been carefully carried out and, reassuringly, the uncertainty budgets provided are consistent with what one would expect for such measurements.

Certainly the work warrants publication.

I would like to have seen a description of the storage and handling of the chemicals.

Response: the details on the storage and handling were added to Sec. 2.1:

“Water content in p-xylene was determined by Fischer titration and equaled 0.01 % (mole fraction). Before the measurements, n-octanophenone and n-octadecanophenone were dried in vacuo to remove any traces of moisture.”

In the text, units are sometimes written using the mid-dot as separator and other times using the space. One of these must be the style of the journal.

Response: in all units, mid-dot has been used as separator.

Reviewer 2 Report

In the presented work the authors continue their investigations of the temperature dependence of the fusion enthalpy of organic non-electrolytes using solution calorimetry (SC). They consider SC to be a useful alternative to heat capacity measurements. As an object of study, n-octadecanophenone, a representative of arylaliphatic compounds, was chosen. The “athermal” solvent - p-xylene -appropriate for evaluating the fusion enthalpy at 298.15 K from the solution enthalpy of crystal was selected. New data obtained by SC and DSC allowed to evaluate independently the correction between ΔcrlH(298.15 K) and ΔcrlH(Tm). It is appeared to be consistent for the two methods validating the used approach.

The paper is well-written and clear. The authors’ assumptions look reasonable. The new experimental data (solution enthalpies, fusion enthalpies, heat capacities) support the authors’ conclusions. Nonetheless, some of the presented data raise questions and have to be commented. This is especially important, given the very small value of the correction between ΔcrlH(298.15 K) and ΔcrlH(Tm), which is comparable to the measurement errors.

1. Analyzing data in Table 1 one can suspect that the fusion enthalpy value depends on the sample mass in direct ratio. This needs to be commented.

2. The heat capacity of solid n-octadecanophenone was measured by DSC at 322.5 K but not directly at target 298.15 K, why so. Please, specify the masses of the n-octadecanophenone samples (line 102). It would be helpful to add in paper the experimental DSC data (for example, as a Figure).

3. The uncertainty of 2 % is ascribed to the extrapolated value Cp,m(l, 322.5 K) = 697 ± 14 J K-1 mol-1 (line 132). Considering a quite large (c.a. 50 degree) extrapolation range this uncertainty has to be apparently extended. Whether it is possible to extend the used range of heat capacity measurements of liquid n-octadecanophenone (373-418 K) to the point of crystallization? This would provide a better basis for extrapolation.

Noticed typos:

Line 50. …solute solvent-systems… probably meant … solute-solvent systems...

Line 54. A colon should replace dot after Tm.

Line 121. Table 2. The enthalpies… probably meant …The solution enthalpies…

Line 133. …range of Eq. (2)… probably meant …range of Eq. (1)…

Author Response

We thank the reviewers for their attention and valuable comments that helped to improve the article. The changes in the manuscript were made in review mode and can be seen as a red underlined text. The responses to each comment are provided in docx file.
